



# Vicarious Calibration of the TROPOMI-SWIR module over the Railroad Valley playa

Tim A. van Kempen [1], Tim J. Rotmans[2,1], Richard M. van Hees [1], Carol Bruegge[3], Dejian Fu [3], Ruud Hoogeveen [1], Thomas J. Pongetti[3], Robert Rosenberg [3], and Ilse Aben [1,4]

[1]SRON Netherlands Institute for Space Research, Niels Bohrweg 4, 2333 CA, Leiden, the Netherlands
[2]TU Delft, Faculty of Aerospace Engineering, Kluyverweg 1, 2629 HS, Delft, the Netherlands
[3]Jet Propulsion Laboratory, California Institute of Technology, 4800 Oak Grove Drive M/S 183-601, Pasadena, CA 91109, United States of America
[4]Vrije Universiteit, Department of Earth Sciences, De Boelelaan 1081, 1081 HV, Amsterdam, the Netherlands

**Correspondence:** Tim van Kempen (t.a.van.kempen@sron.nl)

**Abstract.** The SWIR module of the TROPOMI instrument on board ESA's Sentinel-5p mission has been very stable during its five years in orbit. Calibration was performed on-ground, complemented by measurements during inflight instrument commissioning. The radiometric response and general performance of the SWIR module are monitored by onboard calibration sources. We show that after five years in orbit, TROPOMI-SWIR has continued to show excellent performance with degra-

dation of at most 0.1% in transmission and having lost less than 0.3% of the detector pixels. Independent validation of the instrument calibration, via vicarious calibration, can be done through comparisons with ground-based reflectance data. In this work, measurements at the Railroad Valley Playa are used to perform vicarious calibration of the TROPOMI-SWIR measurements, using both dedicated measurement campaigns, as well as automated reflectance measurements through RADCALNET. As such, TROPOMI-SWIR is an excellent test case to explore the methodology of vicarious calibration applied to infrared

spectroscopy. Using methodology developed for the vicarious calibration of the OCO-2 and GOSAT missions, the absolute radiometry of TROPOMI-SWIR performance is independently verified to be stable down to ∼6-10 % using the Railroad Valley, both on the absolute and, thus, relative radiometric calibration. Differences with the onboard calibration originate from the BRDF effects of the desert surface, the large variety in viewing angles, and the different sizes of footprints of the TROPOMI pixels. However, vicarious calibration is shown to be an additional valuable tool in validating radiance-level performances of

infra-red instruments such as TROPOMI-SWIR in the field of atmospheric composition.





# 1 Introduction

The Tropospheric Monitoring Instrument (TROPOMI[1]) was launched on October 13th, 2017 as the sole instrument onboard the Sentinel-5 precursor (S5p) mission. S5p is part of the Copernicus program headed by the European Commission in partnership with the European Space Agency (ESA). TROPOMI is the first Copernicus instrument aimed at monitoring the chemical

composition of Earth's atmosphere over time (for an overview of the instrument design and performance, including the immersed grating, see e.g.,  Veefkind et al., 2012; Hoogeveen et al., 2013; van Amerongen et al., 2017; Kleipool et al., 2018; van Kempen et al., 2019; Ludewig et al., 2020) The instrument consists of two modules: the UVN module covering ultraviolet (UV), visible (VIS), and near-infrared (NIR) wavelengths and the SWIR module[2] which contains an immersed grating[3] (van Amerongen et al., 2017) and a HgCdTe detector[4] measuring the short-wavelength infrared (SWIR) wavelengths between

2305 and 2385 nm. The SWIR module was specifically designed to measure the dry air columns of carbon monoxide (CO) and methane (CH$_4$). With its swath width of ~2600 km, TROPOMI provides daily global coverage. The detector has 1000 columns in the spectral dimension and 256 rows in the spatial dimension, of which about 975 columns and 217 rows are illuminated. This provides an across-track resolution of 7 km at the nadir and a spectral resolution of 0.25 nm. The along-track resolution is set by the total integration time per frame. This resolution was 7 km at the start of the mission, but adjusted to 5 km just over

one year in nominal operations.

Earth-observing sensors such as TROPOMI are meticulously calibrated before launch to relate their radiometric response to standard Systeme Internationale (SI) units. After launch, most instruments, including TROPOMI, rely on onboard calibration (OBC) systems. OBC systems typically include lamps and/or other calibration targets physically within the spacecraft. For

TROPOMI-SWIR, a white light source, a dedicated LED, a set of five monochromatic diode lasers, and a cold black surface are available. In addition, TROPOMI can measure the solar irradiance. To ensure the reliability of the data products, monitoring of the performance and radiometric response of the SWIR instrument through the OBC is required. Moreover, the applicability of calibration data used in processing the raw instrument data to spectral radiances and irradiances needs to be kept up to date to ensure radiometric accuracy in the presence of expected response changes.

After the first year of nominal operations, the TROPOMI SWIR module was shown to be very stable with little changes to the calibration in comparison to the on-ground results (van Kempen et al., 2019; Ludewig et al., 2020). Instrument monitoring is available online[5] and affirms the continuing excellent performance of the TROPOMI-SWIR module up to the current date

---

[1]TROPOMI is a collaboration between Airbus Defence and Space Netherlands, KNMI, SRON, and TNO, on behalf of NSO and ESA. Airbus Defence and Space Netherlands is the main contractor for the design, building, and testing of the instrument. KNMI and SRON are the principal investigator institutes for the instrument. TROPOMI is funded by the following ministries of the Dutch government: the Ministry of Economic Affairs, the Ministry of Education, Culture and Science, and the Ministry of Infrastructure and the Environment.

[2]developed by SSTL, United Kingdom

[3]developed by SRON

[4]developed by Sofradir, France

[5]http://mps.tropomi.eu and http://www.sron.nl/tropomi-swir-monitoring





of writing.

Validation of the instrument calibration and monitoring from the OBC is, in general, a challenge. For space-based sensors, historically it has been more typical to compare the final data products (i.e. column densities of $CH_4$ and/or CO in the case of TROPOMI-SWIR) to ground-based measurements. However, differences reveal very little about the calibration of individual instruments. Independent validation of the instrument calibration can be performed through routine measurement of Pseudo-Invariant Calibration Sites (PICS), cross-calibration with other instruments, and/or a vicarious calibration from sites with dedicated instrumentation. The first method, radiance measurements of PICS, can relatively accurately monitor instrument degradation (i.e. the relative radiometric calibration) but provides relatively poor results regarding absolute radiometry. For TROPOMI-SWIR this was done using desert sites in the Sahara, Saudi Arabia, and Namibia (van Kempen et al., 2021). The other two methods rely on external measurements, either from complementary space-based sensors (i.e., cross-calibration) or (dedicated) ground-based measurements (i.e., vicarious calibration).

An often-used site for vicarious calibration is the Railroad Valley (RRV) Playa in Nevada, USA (latitude: 38.475 deg, longitude: -115.69 deg). Due to its size (approximately circular with a 12 km diameter) and flat relatively homogeneous surface, RRV is a relatively ideal test site to perform vicarious calibration of the TROPOMI-SWIR module. It also has a unique combination of surface conditions, a high amount of cloud-free days, and accessibility. The site is equipped with permanent monitoring instrumentation such as through the RADCALNET network (Bouvet et al., 2019) or the JPL LED. In addition, dedicated measurement campaigns (e.g. Bruegge et al., 2019a) have been carried out for the OCO-2, OCO-3 and GOSAT missions (e.g. Bruegge et al., 2021). Although similar sites in Nevada, such as the Ivanpah playa and Rogers dry lake, have been explored by other studies, these sites are often smaller and thus not suitable for atmospheric sounders with relatively large footprints, such as TROPOMI, NASA's OCO-2 and OCO-3 and/or JAXA's GOSAT and GOSAT-2 missions. Worldwide, other sites of sufficient size exist, but these are often not homogeneous due to human activity or can be inaccessible due to their remote location, geopolitical circumstances, and/or environmental conditions.

In this paper, we will present the current performance of the TROPOMI-SWIR module after five years in flight, and show vicarious calibration of the module using the RRV Playa. Section 2 shows the TROPOMI data and its performance as monitored by the OBC. Section 3 in turn presents the other data used for vicarious calibration. In Section 4 we discuss the necessary correction and processing steps. Section 5 gives the results of the comparison of TROPOMI-SWIR radiance data with the vicarious calibration reference data sets. Results are first discussed in a broader context in section 6 and in turn summarized in the conclusions in section 7.

## 2 TROPOMI-SWIR Performance

The SWIR (Short Wavelength Infra Red) module of the TROPOMI instrument has a wavelength coverage between 2305 and 2385 nm. Atmospheric transmission in the TROPOMI-SWIR module wavelength range is dominated by strong absorption features of water vapour ($H_2O$) and methane ($CH_4$). In addition, features of both deuterated water (HDO) and carbon monoxide



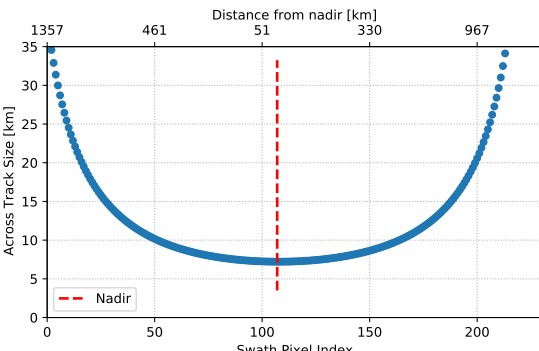

**Figure 1.** Spatial resolution in the across-track direction as a function of swath pixel index and distance from nadir.

(CO) can be identified. Only a very limited wavelength range can be considered to be line-free. For this work, continuum radiance (i.e. line-free) refers to a wavelength range of 2313 to 2313.3 nm.

The SWIR module has a spectral resolution of approximately 0.25 nm and performs soundings at spatial resolutions up to $7 \times 5.5$ km$^2$ (originally $7 \times 7$ km$^2$). A change in the integration time of the scientific operating mode was implemented in August 2020, effectively changing the along-track resolution from 7 km to 5.5 km. The high-inclination (98.7°) near-polar sun-synchronous orbit has an orbital cycle of 227 orbits ($\approx$16 days) and an ascending node equatorial crossing time at 13:30 Mean Local Solar time above nadir. The wide swath of 2600 km allows TROPOMI to provide daily global coverage of Earth's radiance for latitudes outside $\pm 7°$ of the equator. Within the equatorial latitudes, daily coverage is at least 95%.

TROPOMI radiance measurements of any geographical location show significant variations from day to day due to the large swath. As the swath is near to symmetric around a nadir-viewing geometry, both east and west viewing directions are possible; viewing zenith angles have a range from 0 to 66 deg. In effect, the spatial footprint in the across-track direction is a function of the instrument zenith angle (i.e., distance from the nadir pixel). This dependency is shown in Fig. 1. The wide swath impacts the crossing times over RRV in local time, which can vary by +/- 0.8 hours, depending on the location of the measurement in the swath.

Figure 2 shows examples of five different spatial soundings over the central location in RRV. Both the on-ground area's size and orientation vary. Larger pixels include significant portions of the surrounding mountainous area and the orientations of the pixels shift depending on an eastern or western viewing angle. In total, 227 distinct combinations of TROPOMI pixel orientations and shapes cover the central location of RRV.

In addition to the radiance measurements, TROPOMI-SWIR measures the solar irradiance at regular intervals (i.e. every 3-5 orbits). See Kleipool et al. (2018), van Kempen et al. (2019) and Ludewig et al. (2020).



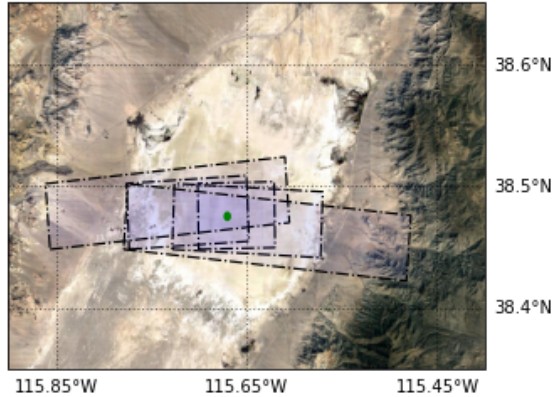

**Figure 2.** TROPOMI pixel shapes over RRV covering central reference location (green ellipse). The Google Tile image (adapted from © Google maps 2019) is shown as the background.

## 2.1 Transmission stability

The instrument calibration of TROPOMI, derived from on-ground measurements, commissioning data, and monitoring data, is described in a set of papers led by the Expert Support Laboratory for the Level-1 processor of the TROPOMI Mission Performance Center (van Hees et al., 2018; Tol et al., 2018; Kleipool et al., 2018; van Kempen et al., 2019; Ludewig et al., 2020). It is stable over the first year. The current instrument stability of the SWIR module, for which independent validation was done in van Kempen et al. (2021), can be found in the SRON online monitoring system[6].

    Figure 3 shows the normalized transmission of the solar irradiances from the start of nominal operations on April 28th, 2018 up to December 18th, 2022. The calibration algorithms and calibration key data were updated in July 2021 in orbit 19258. These changes are accounted for. The fitted trend on the data of the main diffuser shows a slow variation showing an increase of 0.15% within the first two years and a gradual decline to a value 0.1% below the initial value since. This variation is hypothesized to be located in the optical path, as evidenced by the reproduction using the backup diffuser. This stability is corroborated by the data from the on-board White light source (WLS), which shows that the trend seen in both irradiance data streams (<0.15%) is consistent with the observed variation and uncertainty of WLS measurements (0.35%). Uncertainties are given using the biweight spread to remove outliers from the full array. The uncertainties on the numbers given above are small, with individual measurements of the solar irradiance providing uncertainties 0.03%. The uncertainty on individual measurements appears to increase as a function of time. On the WLS, uncertainties on individual measurements are also of this order, although the variance from measurement to measurement is larger. Inspection of individual frames reveals this uncertainty is dominated by the variations and uncertainties of the WLS output signal.

---

[6]www.sron.nl/tropomi-swir-monitoring/





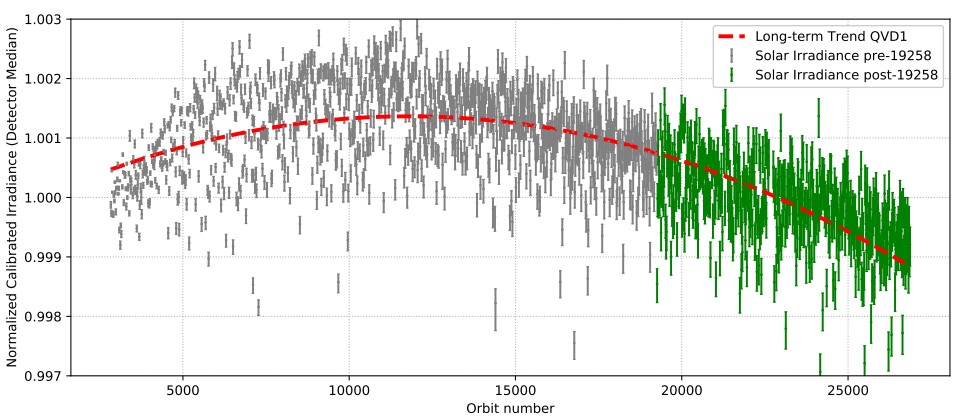

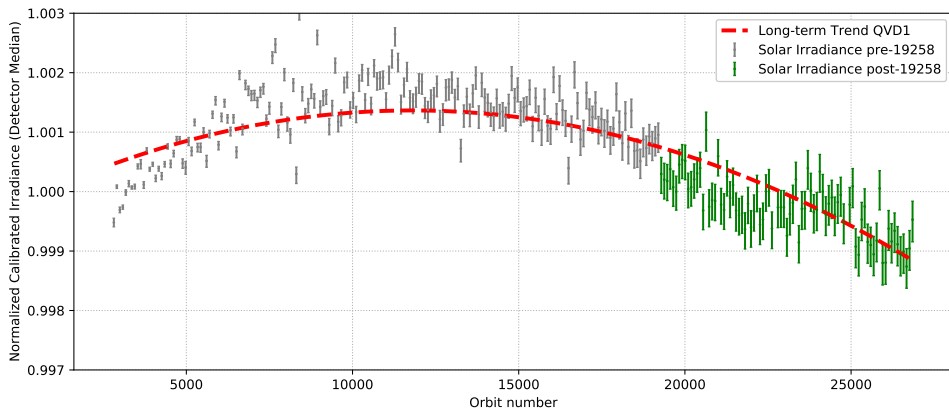

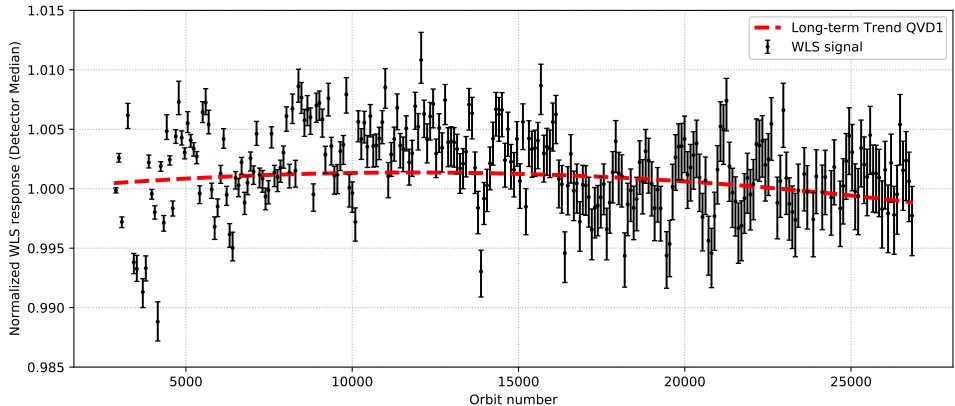

**Figure 3.** Long-term stability of the TROPOMI-SWIR module shown by the detector median of the normalized solar irradiance using the main diffuser (top), back-up diffuser (middle) and White Light Source (bottom). The change in calibration key data for the irradiance is represented in the data before (grey) and after (green) orbit 19258. The red line shows the long-term trends of solar irradiance in all three plots. Orbit number 25000 corresponds to Nov. 2022.



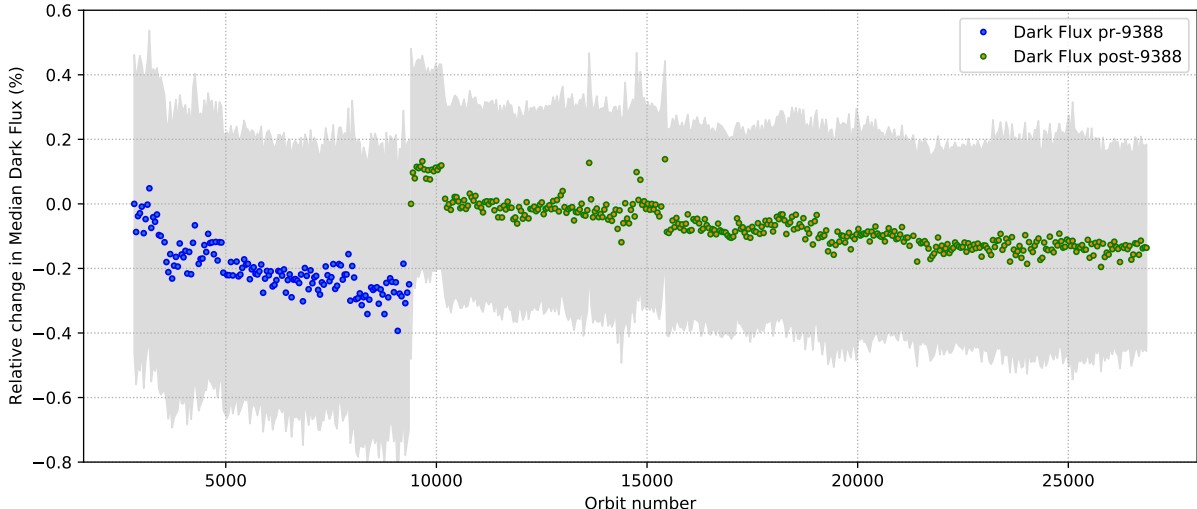

**Figure 4.** Long-term stability of the detector median of the dark flux generated by the TROPOMI-SWIR module normalized at the start of nominal operations (April 2018). Orbit number 25000 occurred during November 2022. The grey colour indicates the uncertainty, derived from the bi-weight spread over the detector.

## 2.2  Dark Flux

The median of the dark flux of the detector is a representation of the overall performance of the detector. If the hardware in this part of the instrument degrades (e.g., electronics, amplifiers) this will be seen in an increase in the median of dark current of the quarter million pixels. Dark flux is derived by background measurements taken during the night side of the S5P orbit with
the folding mirror mechanism (FMM) mechanism closed.

Figure 4 shows the median dark flux, relative to the dark flux measured at the reference orbit taken at the start of nominal operations (orbit 2756). At this time the median dark flux was 3776 $e^-$/s (van Kempen et al., 2019). Each point uses successful background measurements over 15 consecutive orbits. At the change in along-track spatial resolution, attained by reducing the exposure times of radiance measurements, the exposure times of the radiance background measurements were also changed
to match the new configuration. This caused a small jump in the derived dark flux of 0.3%, mainly due to the absence of measurements at exposure times longer than 844 ms. Figure 4 shows little to no change in the overall dark flux of the detector except a very slow (0.1%) reduction in dark flux.

## 2.3  Detector Pixel degradation

In flight, the detector of the TROPOMI-SWIR is continuously exposed to the vacuum environment of its orbit. Although still
partially shielded from deep space, cosmic rays do hit the detector regularly, in particular during crossings of the South Atlantic





Anomaly and the polar regions. The quality of individual pixels is monitored through a pixels operability (response to light), dark current, noise and noise variation responses. The results of these tests are obtained from dedicated calibration measurements taken during the night side of the orbit. Pixel quality is expressed as a number between 0 (completely unresponsive) and 1 (perfect) (van Kempen et al., 2019). Three categories are used. Good pixels, with a value of quality above 0.8, bad pixels,

with a value of quality between 0.1 and 0.8, and worst pixels, with a value of quality below 0.1. Completely unresponsive pixels (i.e. a value of 0.0) are included in the worst pixel category.

Figure 5 presents the number of pixels in the bad and worst categories since the start of nominal operations. Although the number of pixels with bad or worst performance has been steadily rising, the total number of pixels in these categories remains negligible: an increase of 0.4% from 0.85 to 1.25 % over the total detector over five years in orbit. The change in calibration

key data applied during orbit 19358 reveals a small effect in the up-to-then linear rise in bad and worst pixels. The bulk (i.e. > 90%) of pixels that change category do so due to a change in their noise properties. Only a handful of pixels appear to have become non-responsive since the start of nominal operations.

We note that a large (between 80 and 90 %) fraction of pixels in either the bad or worst category recover to a good performance within a typical time scale of a few weeks to two months. This, combined with the random nature of cosmic ray

impacts, is the main origin of the variation seen from orbit to orbit in the pixel degradation and number of bad and worst pixel. The origin of this behaviour is partly understood but beyond the scope of this paper.

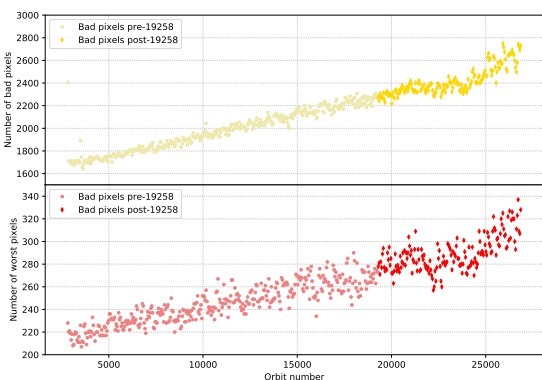

**Figure 5.** Number of detector pixels identified as bad ($0.1 <$ quality $< 0.8$) and worst (quality $<0.1$) over the illuminated part of the detector from the start of nominal operations (April 28th 2018) to the current day. Orbit number 25000 occurred during November 2022. The update of calibration key data during orbit 19358 is indicated with different symbols.

## 2.4 Conclusions on Stability of TROPOMI-SWIR

The main conclusion that can be drawn from the results here is that an extremely (i.e. $\sim 0.1\%$ transmission change and loss of 0.4% of pixels over five years) stable performance of the TROPOMI-SWIR module is seen. TROPOMI-SWIR is thus well-





suited to explore the methodology of vicarious calibration as applied to similar IR spectroscopic sounders. The performance of TROPOMI-SWIR presented here is much better than the limits typically seen for vicarious calibration (Kuze et al., 2014; Bruegge et al., 2019a).

The scale of the degradation found in the sections above is vastly smaller than the typical accuracies achievable using vicarious calibration in studies for comparable sensors to TROPOMI, such as OCO-2 and GOSAT (Kuze et al., 2014; Bruegge et al., 2019b). The scale is also significantly smaller than the limits found for TROPOMI-SWIR instrument degradation by monitoring PICS in the Saharan desert (van Kempen et al., 2021).

## 3 Reference Data

To perform the vicarious calibration, two types of measurements from RRV were used. The first are data extracted from automated instruments within RADCALNET, see 3.1. The second set of measurements consists of dedicated campaign ground measurements, see 3.2. In addition to the reference data, three ancillary datasets were used to derive various necessary corrections, see 3.3. The full method is discussed in 4.

### 3.1 RADCALNET

The Radiometric Calibration Network (RADCALNET) provides zenith-view bottom-of-atmosphere (BOA) and top-of-atmosphere (TOA) reflectance data during daytimes at 30-minute intervals covering wavelengths from 400nm to 2500nm. This is done from five test sites, including RRV (Bouvet et al., 2019). The TOA reflectances are calculated from ground-based zenith-view reflectance measurements using MODTRAN (Berk et al., 2014). Data is publicly available through the RADCALNET website[7]. The RADCALNET dataset for RRV will be used as a reference for the observed TROPOMI continuum signal. RADCALNET is an automated instrument suite and thus can provide data coverage at times during TROPOMI overpasses. Compared to the TROPOMI data, RADCALNET data is limited in its spectral resolution (Bruegge et al., 2021). The hyperspectral data cube has a resolution of 10 nm, approximately 40 times wider than the TROPOMI-SWIR module. In addition, the response of a single spectral bin is triangular. For more on the RADCALNET dataset, see either Bouvet et al. (2019) or the User guide on the website.

### 3.2 RRV campaigns

Dedicated measurements campaigns were held at the RRV site to perform vicarious calibration for the OCO-2, OCO-3 and GOSAT and GOSAT-2 atmospheric sensors (e.g. Kuze et al., 2014; Bruegge et al., 2019b, a). The field measurements of the surface reflectance of the site are done by a field spectrometer manufactured by ASD (now part of Malvern Panalytical) over an area of 500 meters by 500 meters as close in time to the overpass time of the OCO and GOSAT sensors, both of which are also close to the 13:30 mean crossing time of TROPOMI at nadir. The methodology for the field measurements is described in Section IV-A in Bruegge et al. (2019b). The data collection includes the meteorological data from a resident weather station,

---

[7]see https://www.radcalnet.org/




| Orbit | Date | Geometry | | | | Radiance | Multiple |
| | | Solar | | Instrument | | | Measurements |
| | | Zenith | Azimuth | Zenith | Azimuth | [$10^{12}$ photons/s | Available |
| | | [deg] | [deg] | [deg] | [deg] | /nm/sr/cm$^2$] | |
| 3652 | 2018-06-27 | 15.80 | -162.98 | 50.76 | 63.62 | 6.50 | Y |
| 3667 | 2018-06-28 | 26.34 | -118.05 | 53.47 | -91.60 | 7.28 | Y |
| 3681 | 2018-06-29 | 23.15 | -125.40 | 34.53 | -95.91 | 7.88 | N |
| 3695 | 2018-06-30 | 20.40 | -134.52 | 7.14 | -94.26 | 7.84 | Y |
| 3709 | 2018-07-01 | 18.03 | -145.73 | 23.32 | 68.84 | 6.78 | N |
| 3723 | 2018-07-02 | 16.42 | -159.36 | 45.53 | 65.00 | 6.95 | Y |
| 8874 | 2019-06-30 | 29.63 | -112.51 | 65.38 | -87.63 | 5.77 | Y |
| 8887 | 2019-07-01 | 15.44 | -179.49 | 63.05 | 59.50 | 5.28 | Y |
| 8902 | 2019-07-02 | 23.18 | -126.01 | 34.53 | -95.91 | 7.78 | Y |
| 8930 | 2019-07-04 | 18.15 | -146.44 | 23.32 | 68.84 | 6.64 | Y |
| 8944 | 2019-07-05 | 16.58 | -160.02 | 45.54 | 65.01 | 7.02 | Y |
| 14123 | 2020-07-04 | 23.32 | -126.54 | 35.12 | -95.81 | 8.86 | Y |
| 14137 | 2020-07-05 | 20.61 | -135.78 | 7.69 | -94.80 | 8.76 | Y |
| 15272 | 2020-09-23 | 42.57 | -152.01 | 6.59 | -93.63 | 6.05 | N |
| 17996 | 2021-04-03 | 35.88 | -152.34 | 8.24 | -95.25 | 6.93 | N |
| 18932 | 2021-06-08 | 16.77 | -155.86 | 45.54 | 65.01 | 7.56 | Y |
| 19358 | 2021-07-08 | 20.77 | -136.72 | 7.14 | -94.27 | 8.01 | Y |
| 19372 | 2021-07-09 | 18.59 | -148.09 | 23.32 | 68.83 | 7.07 | N |
| 23231 | 2022-04-07 | 33.15 | -158.62 | 22.17 | 68.94 | 6.29 | Y |

**Table 1.** Overview of TROPOMI orbits coinciding with dedicated ground campaigns. The associated geometry of the solar and instrument zenith and azimuth angles is given. In addition, the TROPOMI-SWIR continuum radiance at 2313 nm is provided.

operated by the LSpec network of JPL. Aerosol optical depth (AOD) is retrieved from instruments in the Aerosol Robotic Network (AERONET). Note however that the longest wavelength of this photometer is 1020 nm, significantly outside of the wavelength range of TROPOMI.

5     Table 1 lists the TROPOMI orbits coinciding with dedicated ground campaigns. Some days have multiple measurement points. Table 1 provides the TROPOMI orbit number, date, and geometrical parameters, as well as the observed continuum radiance, measured at 2313 nm. The latter shows that the measured radiance is not constant, varying day to day. This variation is dominated by the varying viewing angles, represented by the Bidirectional Reflection Distribution Function (BRDF) of the non-Lambertian desert surface.




## 3.3 Ancillary data

All data taken during ground campaigns need to be corrected for the non-Lambertian nature of the RRV desert surface. Several methods are discussed (see section 4), for which independent data is needed. These are either the MODIS and VIIRS BRDF data products (see 3.3.1 and 3.3.2) or the MISR-derived mRPV values (see 3.3.3). The MODIS and VIIRS data products (MCD43A1 and VNP43MA1) use a kernel-driven semi-empirical BRDF model, utilizing RossThick-LiSparse-Reciprocal kernel functions for characterizing isotropic ($f_{iso}$), volume ($f_{vol}$) and surface ($f_{geo}$) scattering (Ross, 1981; Li and Strahler, 1992; Wanner et al., 1995; Lucht et al., 2000; Schaaf et al., 2002, 2011). These in turn provide BRDF factors for arbitrary viewing and solar angles. The MISR data product is given as variables of the modified Rahman-Pinte-Verstraete (mRPV) model (Rahman et al., 1993b, a).

### 3.3.1 MODIS

MODIS (Moderate Resolution Imaging RadioSpectrometer) (Justice et al., 1998) consists of two instruments mounted on the AQUA and TERRA satellites. The two instruments are viewing the complete surface of the Earth every 1 to 2 days under multiple angles, simultaneously acquiring data in several spectral bands. The closest band to the TROPOMI-SWIR wavelength range is Band 7 at 2.1 microns. BRDF products provide scattering values $f_{iso}$, $f_{vol}$ and $f_{geo}$, which are used to derive the BRDF at 16-day intervals. It is publicly available with the designation MCD43A1 (See e.g., Schaaf et al., 2011). The MODIS spatial resolution is significantly higher than TROPOMI, with the MCD43A1 product provided at a resolution of 500x500m$^2$. As such it is of higher resolution than the VIIRS product described in the next section. Because of their high albedo at SWIR wavelengths, desert surfaces often saturate MODIS detectors and/or are flagged as fully or potentially cloudy. MODIS data can thus not be fully reliable. The data is provided in a sinusoidal coordinate system with the RRV playa included in the (h,v) = (8,5) grid tile.

### 3.3.2 VIIRS

The VIIRS (Visible Infrared Imaging Radiometer Suite) instrument on the Suomi-NPP (National Polar Partnership) satellite (Justice et al., 2013), launched in 2011. It provides a BRDF product over land. The VIIRS BRDF product (VNP43MA1) provides data at 1 km resolution at daily intervals using observations over a total of 16 total days (Schaaf et al., 2019). It is weighted to the ninth day. The VNP43 product is designed as a continuation of the MODIS product (Liu et al., 2017) and uses the same coordinate system as MODIS. Band M11 between 2.23 and 2.28 microns which is located close to the TROPOMI band was used. Contrary to MODIS, the VIIRS instrument is much less affected by saturation and/or false cloud flagging at these wavelengths.

### 3.3.3 MISR

Data from the Multi-Angle Imaging SpectroRadiometer (MISR) (e.g. Diner et al., 2002; Bruegge et al., 2002) can be used to derive BRDF parameters at a time resolution of 15 days (Bruegge et al., 2002, 2019a). MISR consists of nine sensors that





observe at angles ranging from 70.5 degrees forward to 70.5 backward direction from the local vertical. MISR has four spectral bands at 446, 557, 672, and 862 nm and a surface BRF product with which TOA radiances are corrected for atmospheric effects. Similar to the MODIS sensors, the high reflectivity of RRV and the potential presence of clouds cause the number of data points used as input to derive the values to be irregular. The surface reflectances from MISR are fitted to the mRPV model

described below. The values for the free parameters in the mRPV models were found to be suitable (Bruegge et al., 2019a). The data is freely available in product MIL2ASLS.003 through the atmospheric science data website of NASA[8].

## 4   Correction methodology

To compare data from these three sources (TROPOMI, RadCalNet, dedicated campaigns), several correction steps are required using the ancillary data (VIIRS, MODIS, MISR). Corrections are grouped into three categories.

1. The first category is a correction due to scattering and absorption by aerosol and molecules along the light path through the Earth's atmosphere towards TROPOMI. This is corrected using radiative transfer calculations, which include the optical depth (i.e., the AOD). These are discussed in 4.1

2. The second category covers corrections due to surface properties influencing the reflection on the RRV surface. These are carried out through a normalization factor of the BRDF. For RRV the surface is assumed to be a homogeneous flat

desert area within the full areas covered by individual TROPOMI-SWIR pixels. As such, the BRDF depends on the solar illumination angles ($\theta_{\text{sol}}$, $\phi_{\text{sol}}$) and instrument viewing angles ($\theta_{\text{i}}$, $\phi_{\text{i}}$), given in polar coordinates at the overpass time with the polar coordinates corresponding to a nadir view at $\theta = 0$. The convention for $\phi_{\text{sol}}$ is anticlockwise. This is presented in 4.2

3. The third category of corrections consists of corrections relating to coverage. These are related to time differences

between respective soundings and associated ground measurements, spatial averaging over RRV, and/or different spectral coverages. This is discussed in 4.3.

### 4.1   Radiative Transfer

### 4.1.1   RADCALNET ToA radiative transfer

The RADCALNET product provides a Top of Atmosphere (ToA) product, for which the reflectance value is given with the

atmospheric effects included. This is done using the MODTRAN V5.3 radiative transfer code, with the method and output data described in sections 3.3 and 3.4 of Bouvet et al. (2019). The data has used a representation of the continuum values in the TROPOMI band. A continuum radiance for the RADCALNET data is derived by multiplying the ToA reflectance with the observed TROPOMI-SWIR solar irradiance at 2313 nm.

---

[8]https://asdc.larc.nasa.gov





### 4.1.2 RemoteC

For the comparison between TROPOMI and ground-campaign reflectances, the radiative transfer is calculated to obtain simulated spectra from reflectances measured on the ground. This is done with RemoteC. RemoteC is the retrieval code used for the methane operational product of TROPOMI (Hasekamp and Butz, 2008; Butz et al., 2009, 2011; Hu et al., 2018). For these retrievals, a solar model (Hu et al., 2016) is adopted in addition to a set of standard settings, such as the location and altitude of RRV, the number of atmospheric layers, and the input values of cross sections of relevant chemical compounds (water, carbon monoxide, and methane). For the initial column densities, original MIPREP estimates are used instead of derived TROPOMI abundances. This was done specifically to avoid introducing instrument systematics of the TROPOMI instrument. For more on RemoteC and its usage, we refer the reader to Hu et al. (2016).

### 4.1.3 Aerosol Optical Depth

The Aerosol optical depth (AOD) changes from day to day and needs to be corrected for in the radiative transfer. It is available through both the RADCALNET dataset and the RRV ground measurement campaigns. Although it has a dependency on wavelengths, values at infrared wavelengths are typically very low. An assumption was made that the AOD at 2.3 microns is similar to or lower than the AOD at 1 micron, which is a value obtained from ground measurements using Aeronet. Simulations with an extrapolated linear drop-off from 1 to 2.3 microns showed differences less than $1\%$ in the continuum radiance results.

### 4.2 BRDF normalization

BRDF normalizations are done to correct for effects introduced byt the viewing angles ($\theta_i$, $\phi_i$) of the instrument. Solar illumination after the reflection on the desert surface to angles $\theta$, $\phi$ is expressed as signal $S$.

$$S(\theta_i, \phi_i) = S_{\mathrm{norm}}(0,0)/nBRDF(\theta_i, \phi_i) \tag{1}$$

with $0,0$ representing the nadir direction, $S_{\mathrm{norm}}(0,0)$ the amount of light scattered towards nadir and $nBRDF(\theta_i, \phi_i$ the normalization factor. Several methods can be used to derive $nBRDF$. Two are considered here:

1. A factor derived from an mRPV model (section 4.2.1). These utilize results extracted from the data from the MISR measurements.

2. Satellite measurements of the BRDF itself are used to derive the $nBRDF$ factor. The results from BRDF products of both MODIS (3.3.1) and VIIRS (3.3.2) are considered.

### 4.2.1 mRPV

The modified Rahman-Pinte-Verstraete (mRPV) model (Rahman et al., 1993a, b) has been used to provide model estimates of a wide variety of surfaces. The model is physics-based and includes a description of a hot spot. The latter, originating from the





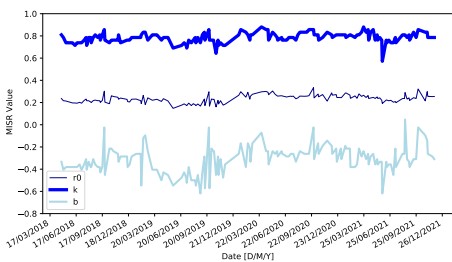

**Figure 6.** MISR values for free parameters in the mRPV model ($r_0$, $k$ and $b$) above RRV as a function of date

radiation transmission through the porous material, was found to be required to accurately describe the increased amounts of observed backward scattering. The mRPV model has been applied to a variety of soil and vegetation surfaces, including RRV (e.g. Bruegge et al., 2019a). The model can mathematically be expressed as

$$BRF(\theta_{\text{sol}}, \theta_{\text{i}}, \phi) = r_0(\cos\theta_{\text{i}}\cos\theta_{\text{sol}}(\cos\theta_{\text{i}} + \cos\theta_{\text{sol}}))^{k-1}$$
$$h\exp(-b\cos\xi) \tag{2}$$

where

$$h(\theta_{\text{sol}}, \theta_{\text{i}}, \phi) = 1 + (1 - r_0)/(1 + G(\theta_{\text{sol}}, \theta_{\text{i}}, \phi)) \tag{3}$$

,

$$G(\theta_{\text{sol}}, \theta_{\text{i}}, \phi) = \sqrt{\tan^2\theta_{\text{i}} + \tan^2\theta_{\text{sol}} - 2\tan\theta_{\text{i}}\tan\theta_{\text{sol}}\cos\phi} \tag{4}$$

,

$$\cos\xi = \cos\theta_{\text{i}}\cos\theta_{\text{sol}} + \sin\theta_{\text{i}}\sin\theta_{\text{sol}}\cos\phi \tag{5}$$

and

$$\phi = \phi_{\text{sol}} - \phi_{\text{i}} \tag{6}$$

In this model, solar and viewing angles $\theta_{\text{sol}}$, $\phi_{\text{sol}}$, $\theta_{\text{i}}$, $\phi_{\text{i}}$ are used in combination with free parameters $r_0$, $k$ and $b$. The factor $h$ relates to the hot spot.

The $nBRDF$ factor itself is derived by dividing $BRF(\theta_{\text{sol}}, \theta_{\text{i}}, \phi)$ by $BRF(\theta_{\text{sol}}, 0, \phi)$ with $\phi_{\text{i}} = 0$. For this nadir view, equations 2 to 6 reduce to

$$BRF(\theta_{\text{sol}}) = r_0(\cos\theta_{\text{sol}}(1 + \cos\theta_{\text{sol}}))^{k-1} \qquad h\exp(-b\cos\theta_{\text{sol}}) \tag{7}$$



with

$$h(\theta_{\mathrm{sol}}) = 1 + (1 - r_0)/(1 + \tan\theta_{\mathrm{sol}}) \tag{8}$$

I.e., the sole angular parameter being the solar zenith angle $\theta_{\mathrm{sol}}$.

Bruegge et al. (2019a) explored the validity of this method as applied to the RRV site, making comparisons with the results
of various ground-based measurement campaigns and satellite measurements. This included derivations for the values of $r_0$,
$k$, and $b$. It was concluded that these appear to vary on levels of a few percent in both time and exact location within the RRV
playa. The wavelength dependency of these parameters (i.e. the similarity of data at short and long wavelengths) appears to be
smaller than typical measurement errors (see e.g. Fig. 7 in Bruegge et al. (2019a)).

Figure 6 shows these values for the RRV central location. Although only the central square kilometer is used, it is assumed
that the values are valid (i.e. continuous and similar) for the entire RRV playa. It shows that most variations in these values are
time variations presumably due to the changes in the surface.

### 4.2.2 BRDF products

The normalization factor can also be derived directly from albedo products from MODIS and VIIRS with the $nBRDF$ factor
expressed as:

$$nBRDF(\theta_{\mathrm{i}}\,\phi_{\mathrm{i}}) = BRDF_{instrument}(\theta_{\mathrm{i}}\,\phi_{\mathrm{i}})$$

$$/BRDF_{instrument}(0,0) \tag{9}$$

Here, $BRDF_{instrument}$ is derived from the scattering parameters of the products of VIIRS and MODIS, i.e., the isotropic
($f_{iso}$), volume ($f_{vol}$) and surface ($f_{geo}$) weighting parameters. These are associated with a RossThickLiSparseReciprocal
BRDF model. The M7 (MODIS, 2.105 to 2.155 micron) and M11 (VIIRS, 2.23 to 2.28 micron) bands were chosen for their
proximity to the TROPOMI-SWIR band. Potential differences in wavelength of these products are assumed to be small. The
obtain the proper values at the TROPOMI geometries, BRDFs are calculated by the forward model. For the calculation, we
refer to the user guides of these products[9].

### 4.3 Coverage Corrections

### 4.3.1 Spatial Averaging

Most data is available at much higher spatial resolutions ($\sim$1 km$^2$ or smaller) than the TROPOMI pixels, which themselves
vary in size. As such, the following assumptions need to be made to ensure a relatively fair comparison.

    1. The RRV playa is homogeneous in its surface properties as measured from the location of the ground measurements.

---

[9]https://www.umb.edu/spectralmass/faculty_staff/crystal_schaaf



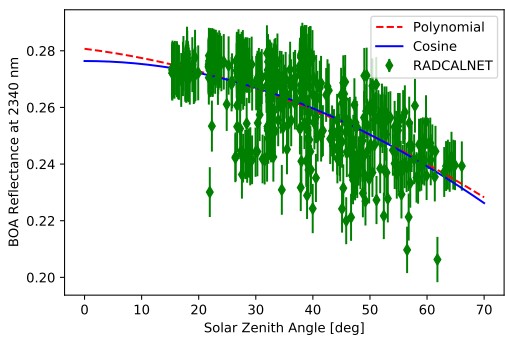

**Figure 7.** Solar zenith angle difference (i.e. time difference) correction using BOA RADCALNET data.

2. If available and included in the inner 5x5 km$^2$ area of the TROPOMI pixel around the reference point of the RRV area, a median of data points will be used. This applies to the MODIS and VIIRS data.

3. More extreme instrument Zenith angles of TROPOMI should be excluded. These pixels include large areas outside of the RRV playa. The limit is set to a 20 km pixel size in the across-track direction (see Fig. 1). This is equivalent to a viewing angle not greater than 60 degrees.

We realize this introduces additional uncertainties to the comparison. These cannot be easily corrected and are expected to be the dominant source of error in the resulting comparison.

### 4.3.2 Time differences and Solar angle

Variations also exist as a function of the time of day. Assuming a homogeneity of a flat desert surface, variations in local overpass time are secondary variations as a function of solar zenith angle that are not accounted for in the individual nBRDF corrections. These need to be corrected when comparing measurements at different times. The variation of the TROPOMI early afternoon overpass can itself span almost 90 minutes in local time. Effectively this is a variation of up to 45 minutes to the RADCALNET data and up to 75 minutes with the RRV ground campaigns. It is assumed that a dependency on time is in effect a dependency on the solar zenith angle.

A time correction function can be derived using the RADCALNET data. With its year-round coverage every 30 minutes during the daytime, one can sample the full range of solar zenith angles and compare the Bottom of Atmosphere results. Averaged BoA reflectances, which are not affected by molecular absorption, between 2310 and 2370 are compared to the measured SZA in bins of 10 nm. Due to the lack of atmospheric absorption, these measurements are more reliable than the top-of-atmosphere values used in the comparison with TROPOMI radiances. Figure 7 shows the fitted dependency, both using a 2nd-degree polynomial fit and a cosine fit is shown. Erroneous data is removed by flagging all values above 0.28 and below 0.20. The cosine fit was found to provide the lowest residuals. The function parameters are 0.276 for the amplitude and 725 as the effective period expressed in units of solar zenith angle. We assume an uncertainty of 10% in these values. A time




correction factor is subsequently derived by calculating the ratio of this cosine function at the two measured solar zenith angles. The correction is smaller than the nBRDF correction, but can at times be the cause of differences of a few percent, due to the range of time differences (up to 1.5 hours) between different TROPOMI measurements that can cover the RRV site.

## 4.4 Spectral Coverages and Resolutions

The spectral coverage of the various instrument differs from that of TROPOMI and must be considered before applying corrections.

### 4.4.1 RADCALNET

RADCALNET data is published at intervals of 10 nanometers, with a triangular-shaped instrument spectral response with a half-width of 10 nanometers. This significant difference as compared to the TROPOMI ISRF of approximately 0.25 nanometers

needs to be accounted for. Near the reference continuum wavelength of 2313 nanometers, several atmospheric features are present that are included in the bandwidth of the RADCALNET point of 2310 nanometers. These are caused by the absorption of water and methane and range in scale of about 2% to almost 80% of the TOA signal. See Figure 8 for a comparison between the TROPOMI-SWIR spectrum as well as the BoA and ToA responses of the RADCALNET data using the average of cloud-free days in June and July 2018. For comparison, the half-base width of the triangular response is shown as a dashed

line.

Using near-nadir $(+/-3\,\mathrm{deg})$ overpass views of TROPOMI above RRV, one can quantify the amount of relative absorption in the RADCALNET signal at 2310 nanometers compared to the TROPOMI continuum value at 2313 nanometers. This value will change due to the varying amount of water and to a lesser extent methane in the atmosphere. This is found to be 24.7% +/- 1.2%. As such a spectral correction multiplication factor of 1.247 will be applied to compare the MODTRAN ToA values

of RADCALNET to the observed 2313 continuum value of TROPOMI-SWIR.

Interpolation of the RADCALNET data down to 1 nm has been performed using its reflectance values (e.g. Bruegge et al., 2021). This strategy was not adopted to the potential uncertainty as compared to the reference field data (see e.g., Fig. 2 of Bruegge et al. (2021) and references therein).

### 4.4.2 Ground campaigns

The ground campaign data are given as reflectances. Radiances are derived by carrying out the radiative transfer as described in section 4.1.2. We refer to Bruegge et al. (2019a) for a description of how these data were obtained and derived. The spectral resolution is much better, approaching the TROPOMI-SWIR module resolution. No additional correction was implemented.

### 4.4.3 Data from ancillary space sensors

The three space sensors used to derive correction factors use various spectral windows. A precise description of the spectral

windows and/or retrieval methodologies can be found in the various user guides. Molecular absorption is avoided in the selec-





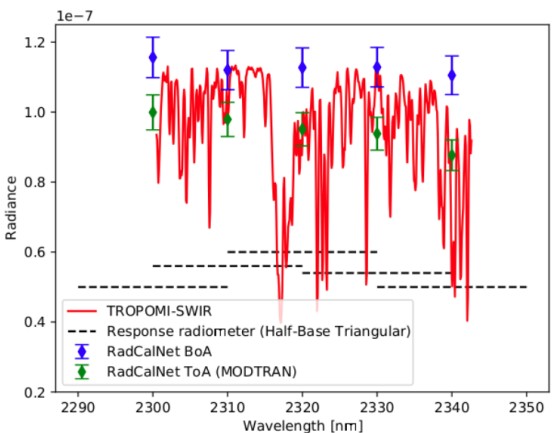

**Figure 8.** Comparison between ToA and BoA data of RADCALNET data with a TROPOMI-SWIR nadir spectrum.

| Orbit | Date | IZA | nBRDF | | |
|---|---|---|---|---|---|
| | | | mRPV | MODIS | VIIRS |
| 3652 | 2018-06-27 | -50.76 | 0.816 | 0.261 | 0.410 |
| 3667 | 2018-06-28 | 53.47 | 1.106 | 0.891 | 0.921 |
| 3681 | 2018-06-29 | 34.53 | 1.105 | 1.151 | 1.116 |
| 3695 | 2018-06-30 | 7.14 | 1.037 | 1.089 | 1.072 |
| 3709 | 2018-07-01 | -23.32 | 0.886 | 0.646 | 0.709 |
| 3723 | 2018-07-02 | -45.53 | 0.806 | 0.344 | 0.457 |
| 8874 | 2019-06-30 | 65.38 | 1.141 | 0.746 | 0.799 |
| 8887 | 2019-07-01 | -63.05 | 0.781 | 0.107 | 0.228 |
| 8902 | 2019-07-02 | 34.53 | 1.106 | 1.147 | 1.120 |
| 8930 | 2019-07-04 | -23.32 | 0.874 | 0.647 | 0.702 |
| 8944 | 2019-07-05 | -45.54 | 0.790 | 0.352 | 0.444 |
| 14123 | 2020-07-04 | 35.12 | 1.084 | 1.078 | 1.069 |
| 14137 | 2020-07-05 | 7.69 | 1.035 | 1.055 | 1.052 |
| 15272 | 2020-09-23 | 6.59 | 1.010 | 1.018 | 1.025 |
| 17996 | 2021-04-03 | 8.24 | 1.027 | 1.030 | 1.032 |
| 18932 | 2021-06-08 | -45.54 | 0.822 | 0.464 | 0.588 |
| 19358 | 2021-07-08 | 7.14 | 1.038 | 1.039 | 1.041 |
| 19372 | 2021-07-09 | -23.32 | 0.873 | 0.838 | 0.832 |
| 23231 | 2022-04-07 | -22.17 | 0.915 | 0.901 | 0.877 |

**Table 2.** BRDF normalization factors derived using the mRPV model and the MODIS and VIIRS BRDF product for the orbits in 1. The TROPOMI-SWIR viewing zenith angle is given as a reference. A positive zenith angle corresponds to a western viewpoint.




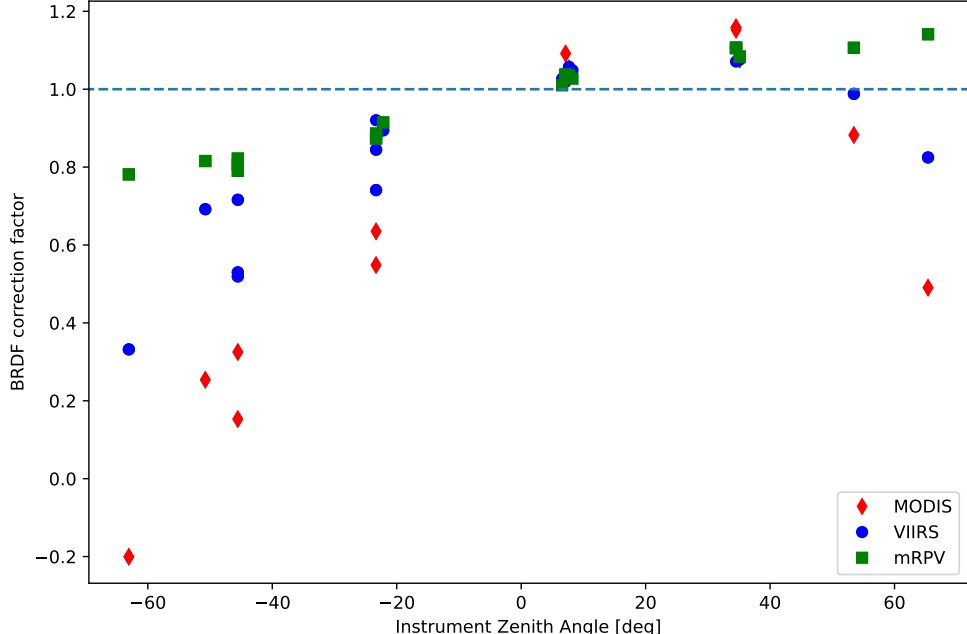

**Figure 9.** Comparison between the nBRDF factors derived from MODIS data (red triangle), VIIRS data (blue circle), or mRPV models (green squares) for the sample, plotted as a function of the instrument zenith angle. With the solar azimuth in the west during the TROPOMI midday overpass, the positive instrument zenith angles coincide with a western azimuth angle of TROPOMI-SWIR.

tion of the spectral window and/or accounted for in the retrieval of the BRDF. Figure 7 of Bruegge et al. (2019a) reveals the wavelength dependency of the $nBRDF$ factor itself to be minimal. This result was obtained by examining various viewing angles at both visible and infrared wavelengths. These results do include the large viewing angles (>40 degrees) not typically found in other atmospheric sensing sounders besides TROPOMI. Bruegge et al. (2019a) implies that the angle dependency of the correction using the mRPV model and MISR data can be considered to be wavelength-independent. Thus, we opted to use results from the closest bands to the TROPOMI-SWIR band, but ignore any differences in wavelength. For MISR the 866-nanometer results are used, while for VIIRS and MODIS the M11 (2.2 microns) and M7 (2.1 microns) bands provided the data closest in wavelength coverage to TROPOMI.





## 5 Results

### 5.1 BRDF correction

The most significant correction that needs to be carried out to compare TROPOMI-SWIR radiances with equivalent values derived from ground-based measurements is the correction for the non-Lambertian surface, i.e. the BRDF correction, $nBRDF$.

Two methods, using three distinct datasets, were discussed in Section 4. Before we inspect the results, it is useful to inspect the differences between the corrections. Table 2 lists the correction factor for each method (i.e., mRPV, MODIS and VIIRS) for the orbital parameters given in Table 1 as well as the instrument zenith angle. Fig. 9 presents the same set of derived correction factors as a function of instrument viewing angle. The difference between the two methods using the BRDF products of MODIS and VIIRS appears to be superficial. Different spatial coverage and data filtering in the two BRDF products was found to be

the origin of differences in these methods. However, there are significant differences between the two BRDF products and the theoretical mRPV model. At larger zenith angles (> 40 deg) the mRPV model and the results from the VIIRS/MODIS method start to significantly deviate, both from an eastern and western viewing point. The source of this deviation must be either the data quality of the VIIRS and/or MODIS data for more extreme angles, or the accuracy of the mRPV model. The data is inconclusive. Given the physical viewing angles of the BRDF products (e.g. the most extreme angle for MODIS is +/-

49.5deg), BRDF data for MODIS and VIIRS is extrapolated at these angles. MISR does extend to these extreme angles (70.5 deg). We also note that the MODIS and VIIRS data show significantly more scatter than the mRPV model for orbits with similar viewing angles at different periods. Near a zenith of zero (i.e. when the TROPOMI overpass is close to the nadir. This is best seen in the western view at +8 deg IZA), the factors are in agreement. Differences in BRDF were explored for RRV in Bruegge et al. (2019a) and we refer the reader there for more information.

### 5.2 TROPOMI-SWIR vs RadCalnet

Figure 10 shows the continuum radiances of TROPOMI-SWIR and the RADCALNET measurements for two scenarios. In each scenario, we correct the TROPOMI-SWIR radiance to a nadir view, assuming the desert surface of RRV to be a Lambertian reflector (top), as well as assuming the desert surface to be an mRPV model with parameters measured by the MISR data (bottom). Correction using either MODIS or VIIRS was found to be very poor, due to the lack of coincidental coverages.

In addition to the BRDF correction, the correction for the time difference between the TROPOMI-SWIR overpass and the closest RADCALNET data point is applied. For both figures, the relative residual is given in green. The limit for cloud cover is extracted from the VIIRS cloud cover product included in the TROPOMI-SWIR methane product and is set at 0.9. Although this limit appears to be uncharacteristically high, it was discovered that at SWIR wavelengths, the cloud cover algorithm for TROPOMI may be inaccurate over bright desert surfaces such as those found at RRV. The algorithm often erroneously iden-

tifies the high albedo as partly or fully cloudy. Although overcast conditions are easily identified, partly cloudy conditions are known to be particularly difficult to discern. This filter is combined with RADCALNET. This data shows fill values for (too) cloudy conditions. For simplicity, it can be assumed that valid RADCALNET values correspond to relatively clear conditions.



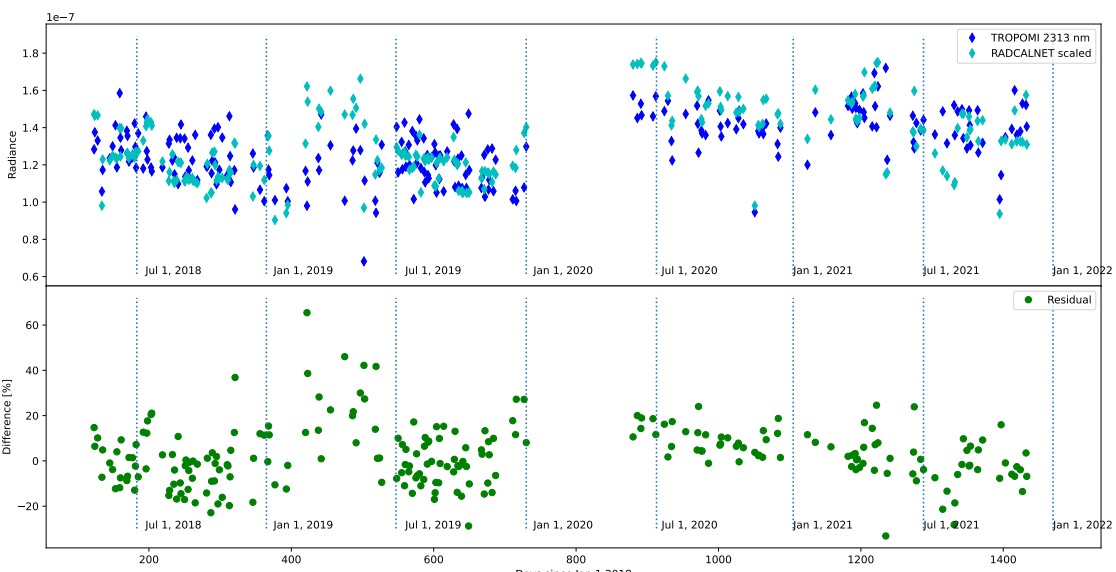

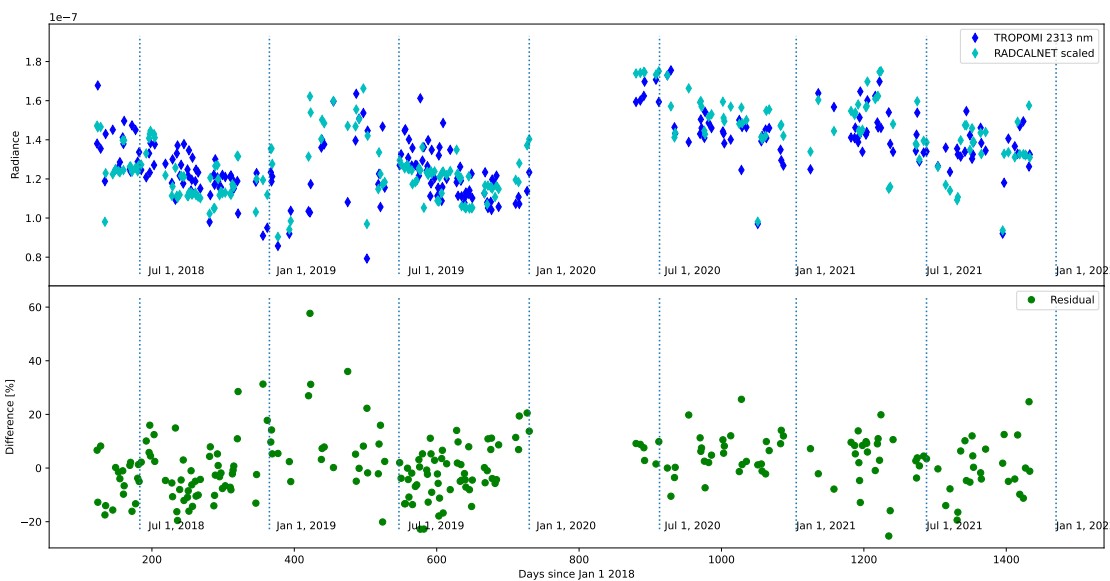

**Figure 10.** Comparison between radiance at 2.3 microns between RadCalNet (cyan) and TROPOMI-SWIR (blue). Assumptions of a Lambertian surface are shown in the top row and an mRPV model at the bottom row.



Several conclusions can be drawn from the trends seen in Fig. 10. First, the day-to-day conformity is relatively poor with clear differences of up to 20% being fairly typical. There is significant scatter in this conformity. Structural trends in the residuals are seen for the Lambertian model, but not the mRPV model. Yearly trends (e.g. a slow reduction in radiance from mid-July to January, and an increase towards early spring) are seen in both datasets. The mRPV model improves the comparison
between the RADCALNET data and the TROPOMI-SWIR data. The median of the absolute difference is 9.97% assuming a Lambertian surface, while this number is reduced to 8.09 % when an mRPV model is used. As expected, neither comparison shows an in- or decreasing residual over time, which would be indicative of instrument degradation. It must thus be assumed the mRPV model is a slightly better representation of the desert surface. However, it must also be assumed that the comparison between RADCALNET and TROPOMI, albeit continuous and simplistic, has severe limitations, which will be discussed in
section 6.

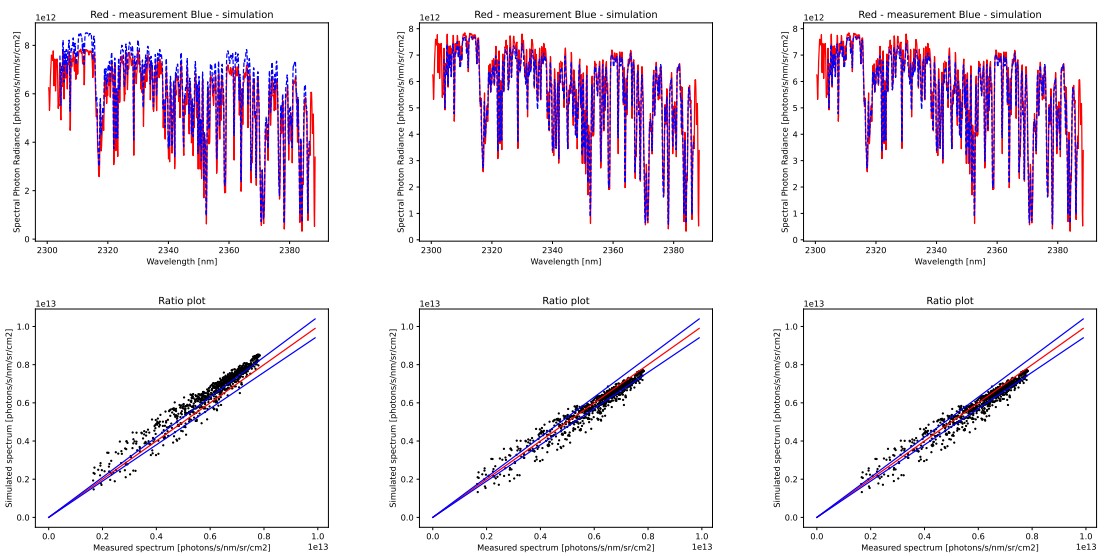

**Figure 11.** Top Row: Comparison between spectra of TROPOMI-SWIR result of orbit 8902 between the observed (red) and simulated (blue). The correction applied is the mRPV model (left), MODIS data (middle), and VIIRS data (right). Bottom row: Plots comparing the radiances of the simulated and observed radiances (black dots) and the one-to-one ratio (red line) and the five-percentile deviations (blue lines) as reference.

### 5.3   TROPOMI-SWIR vs dedicated campaigns

The dedicated ground campaigns are of much higher data quality than that of RADCALNET (For information on data acquisition, see  Bruegge et al., 2019a). The large drawback of using ground campaign data is the coverage. Albeit TROPOMI has daily global coverage, data is limited to a few dedicated days due to the necessary resources to obtain these detailed mea-
surements. For this vicarious calibration, all corrections detailed in section 4 are applied. Figure 11 shows examples of the



comparisons of individual TROPOMI spectra with three spectra simulated from the measured reflectances. The top row shows
TROPOMI-SWIR results in red and simulated spectra in blue. The bottom row shows the difference ratios, where a 1:1 ratio is
given as a red line and a five-percentile difference (both positive and negative) ratio is shown as dashed blue lines. As expected,
the largest deviations within every single comparison are found within deeper methane and water absorption features. This
originates from inaccuracies in the radiative transfer modelling of atmospheric chemical compounds using MIPREP column
densities.

Quantification of the comparison (i.e. finding systematic differences) is obtained by fitting the ratio distributions in the
bottom row of Fig. 11 with a 'constrained' linear relation. For this relation, the intercept is constrained to 0.0 (i.e. no signal in
the observed spectrum equals no simulated signal) instead of using it as a free parameter. In addition, the data range used for
fitting is limited to the top 25% of the TROPOMI-SWIR measured radiance values. This is done to avoid the inaccuracies found
in the deep absorption features. This fitted slope is a quantification of the accuracy of the simulation: a value of 1 equals the
best representation. This constrained linear fit was found to be the most robust as a quantification of the vicarious calibration.
Several other fits and/or comparisons were explored (e.g., averaging of the full spectrum, non-linear fits, continuum-only
comparison, usage of non-zero intercepts), but found to be much less robust. Derived ratios for individual orbits are given in
Table 3. Ratios are presented for each ground measurement. The ratio is also derived for all three correction methods. Figure 12
gives the derived slopes of the ratio's as a function of time (top) and absolute instrument zenith angle (bottom) of TROPOMI
during overpass for all three methods (mRPV, left; MODIS, middle; VIIRS, right).

Ratios above 1.5 have been omitted from the plots in Fig. 12, and are indicated with 'F' in table 3. These do not occur for
the mRPV model, but 14 times for MODIS and 10 times for VIIRS. All dates in which VIIRS ratios are large also show large
MODIS ratios. E.g. orbit 8944 produces ratios over 2 (VIIRS) or even 7 (MODIS), while the mRPV is just 20% lower than the
TROPOMI radiances. Orbit 3709 could not be simulated using RemoteC. The origin of this deviation of VIIRS and MODIS
correction is unknown, but we hypothesize it does not have a singular origin. There does appear to be a correlation with data
taken at large angles, where the inclusion of non-RRV surfaces in the TROPOMI pixel is a factor. In these cases, larger values
are seen in all three ratios. Examples are orbit 8874 or even 15272. In instances where the data for the mRPV model is in good
agreement deviations are thought to be caused by the data coverage and/or quality of MODIS and/or VIIRS BRDF products.
Examples are orbit 8944 or 18932. Large differences between MODIS and VIIRS themselves, indicate data quality limitations
in the BRDF product. Due to its age and problems with bright desert surfaces, it is assumed this disproportionally affects
MODIS.

Table 4 gives the median ratio, average ratio and the median of the absolute deviation. These show that the mRPV model
appears to be significantly better fitted for the median absolute deviation: 6.1% for the mRPV model, as opposed to 11.9 % and
15.5% for the VIIRS and MODIS comparisons respectively. But the median and average ratio slightly favour the MODIS and
VIIRS. We should note that the more extreme ratios are not included. If these are included, both the median and average ratios
for the MODIS and VIIRS methods rise significantly. Inspection of the values does show that the mRPV model has a slight
bias towards a positive ratio (i.e. overestimating the radiance derived from the ground measurements). Due to the presence of
a significant number of extreme outliers and a significantly higher median of the absolute deviation seen for both MODIS and





VIIRS, we conclude that the mRPV model provides a better calibration than the MODIS and VIIRS models. Similar to the results using RADCALNET measurements, the values derived with these methods are an order of magnitude larger than found with the internal calibration units, presented in section 2.

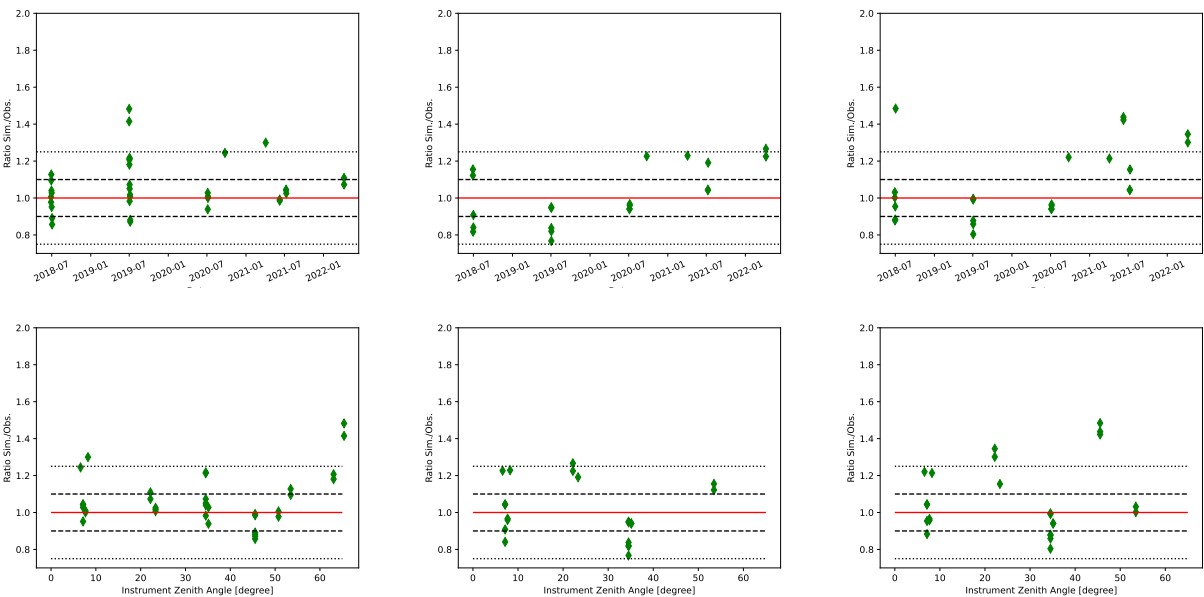

**Figure 12.** Ratio fits of the mRPV (left), MODIS (middle) and VIIRS (left) correction methods as a function of time (top row) and instrument zenith angle (bottom row). Ratios above 1.5 have been omitted.

## 6 Discussion

### 6.1 Degradation constraints from vicarious calibration

The lack of systematics in the residuals of the mRPV model in the RADCALNET comparison (see Fig. 10) places an upper limit to the degradation of TROPOMI-SWIR of 8%. When using a Lambertian assumption, the comparison clearly shows systematic trends, while the upper limit is 10%. The comparison in Fig. 10 between the mRPV model and that of a Lambertian assumption thus reveals correction for the non-Lambertian effects to be a necessary ingredient.

All results presented above, constrain the degradation of the instrument performance over the five years in orbit. However, the limits that were found (i.e., 8% for RADCALNET, 6% for mRPV, 11.9% for VIIRS, 15.5 % for MODIS) are an order of magnitude or larger than the constraints found using the solar irradiance measurements using both on-board diffusers (0.15%). Even limits derived from the WLS measurements (0.5 %) cannot be observed using the vicarious calibration methods.



| Orbit nr | Date | Ratio | | |
|---|---|---|---|---|
| | | mRPV | MODIS | VIIRS |
| 3652 | 2018-06-27 | 0.98 | 4.73 F | 1.73 F |
| 3652 | | 1.01 | 4.87 F | 1.78 F |
| 3667 | 2018-06-28 | 1.1 | 1.12 | 1.0 |
| 3667 | | 1.13 | 1.15 | 1.03 |
| 3681 | 2018-06-29 | 1.04 | 0.82 | 0.88 |
| 3695 | 2018-06-30 | 1.03 | 0.91 | 0.95 |
| 3695 | | 0.95 | 0.84 | 0.88 |
| 3709 | 2018-07-01 | N.A. | N.A. | N.A |
| 3723 | 2018-07-02 | 0.86 | 3.27 F | 1.48 |
| 3723 | | 0.89 | 3.41 F | 1.54 F |
| 8874 | 2019-06-30 | 1.41 | 2.53 F | 1.5 |
| 8874 | | 1.48 | 2.65 F | 1.57 F |
| 8887 | 2019-07-01 | 1.18 | 1.51 F | 4.55 F |
| 8887 | | 1.21 | 1.54 F | 4.65 F |
| 8902 | 2019-07-02 | 0.98 | 0.77 | 0.8 |
| 8902 | | 1.21 | 0.95 | 0.99 |
| 8902 | | 1.05 | 0.82 | 0.86 |
| 8902 | | 1.07 | 0.84 | 0.88 |
| 8902 | | 1.22 | 0.95 | 1.0 |
| 8930 | 2019-07-04 | 1.01 | 2.1 F | 1.55 F |
| 8930 | | 1.02 | 2.12 F | 1.57 F |
| 8944 | 2019-07-05 | 0.88 | 7.34 F | 2.15 F |
| 8944 | | 0.87 | 7.25 F | 2.12 F |
| 14123 | 2020-07-04 | 1.03 | 0.94 | 0.94 |
| 14123 | | 0.94 | 0.94 | 0.94 |
| 14137 | 2020-07-05 | 1.01 | 0.97 | 0.97 |
| 14137 | | 1.0 | 0.96 | 0.96 |
| 15272 | 2020-09-23 | 1.24 | 1.23 | 1.22 |
| 17996 | 2021-04-03 | 1.3 | 1.23 | 1.21 |
| 18932 | 2021-06-08 | 0.99 | 1.54 F | 1.42 |
| 18932 | | 0.99 | 1.56 F | 1.44 |
| 19358 | 2021-07-08 | 1.04 | 1.04 | 1.04 |
| 19358 | | 1.04 | 1.04 | 1.04 |
| 19372 | 2021-07-09 | 1.03 | 1.19 | 1.15 |
| 23231 | 2022-04-07 | 1.11 | 1.27 | 1.35 |
| 23231 | | 1.07 | 1.23 | 1.3 |

**Table 3.** Results from the ratio analysis between the TROPOMI and simulated spectra.





| Method | Med. Abs. Dev. [%] | Median Ratio | Average Ratio |
|--------|--------------------|--------------|---------------|
| mRPV | 6.1 | 1.025 | 1.032 |
| MODIS | 15.5 | 0.950 | 0.996 |
| VIIRS | 11.9 | 0.994 | 1.078 |

**Table 4.** Ratio of the comparisons over the full sample between TROPOMI radiances and the RRV ground campaigns.

These results thus strongly corroborate conclusions made in earlier works on IR atmospheric spectroscopic sounders (for TROPOMI, van Kempen et al. (2021), for OCO-2 Bruegge et al. (2019b), for GOSAT Kuze et al. (2014)). Instrument monitoring and calibration are much more accurate using internal calibration sources than using vicarious calibration sources.

## 6.2 Limits on the vicarious calibration methodology

The important conclusions of this work are not the derived upper limits to the degradation of the instrument found in the vicarious calibration methods. Instead, it is the limits of applicability of this method to push-broom type SWIR sounders for atmospheric composition. At first sight, it appears that vicarious calibration is a severely limiting method. Numerous corrections are required to account for the many uncertainties in this comparison, each introducing new uncertainties. The achieved accuracy can at best be considered acceptable but can be used to find severe deviations in instrument performance. However,

many of these limits are strongly influenced by parameters from the available hardware of this generation of instruments. The following parameters and/or assumptions all affect accurate vicarious calibration using RRV for TROPOMI-SWIR:

- The size of TROPOMI-SWIR pixels.

- The varying size and orientation, and thus the varying coverage, of TROPOMI pixels.

- The time difference between different datasets.

- The non-Lambertian nature of the RRV desert surface and its variation on long time-scales(i.e. the BRDF correction and its variation on the time-scale of months. See e.g. the MISR results).

- The amount and accuracy of cloud filtering.

From this, we conclude that the assumption of homogeneity of the desert surface across RRV (i.e. the reflectance within a TROPOMI pixel) introduces significant uncertainty, and thus limitations, in the vicarious calibration when using RRV as

a reference site. The data from RADCALNET, which represent the reflectance only at a single point, is particularly heavily affected.





### 6.2.1 Cloud filtering

One of the most uncertain aspects in the analysis is the accuracy of the cloud filter. The cloud filter in the TROPOMI methane product is either too strict, and/or has interpreted reflectances of desert surface as partially cloudy when no clouds are present. Two assumptions were made. First, that data availability of the RADCALNET indicates relatively cloud-free soundings. Second, that all data taken during dedicated campaigns are taken on completely cloud-free days. In particular the first assumption may cause additional spread in the comparison between TROPOM-SWIR and RADCALNET. Although a measurement for RADCALNET may be cloud-free in its nadir, clouds may exists over more remote parts of RRV and/or its vicinity. The latter is applicable for larger TROPOMI pixels. Due to the albedo of cloud, TROPOMI-SWIR radiance may be different than estimated from the RADCALNET. Something similar may affect the dedicated ground campaigns, although the level of cloudiness is likely much less due to human element in the data acquisition (i.e. no data acquisition is done during cloudy days) Strict and accurate cloud filtering is of vital importance for a higher confidence in the vicarious calibration, in particular that of RADCALNET.

### 6.3 Implications for TROPOMI

For TROPOMI-SWIR, vicarious calibration using RRV cannot be improved upon from the analysis presented in this work. The main reason for this is that the pixel size of TROPOMI-SWIR is relatively large. Even at nadir, this is 7 by 5.5 km$^2$. But an equally significant influence, and one more unique to TROPOMI-SWIR as opposed to OCO-2, OCO-3, GOSAT and GOSAT-2, is the variation in pixel size and the accompanying orientation, as seen in Fig. 2. To achieve daily near-global coverage, TROPOMI-SWIR has 227 unique and distinct pixel sizes and coverages over RRV. The assumption that the complete area within all of these TROPOMI-SWIR coverages is homogeneous and represented by ground and/or BRDF measurements near (in the case of ground measurements) or at (in the case of RADCALNET) the central location introduces large uncertainties. Measurements at larger viewing angles, i.e. where the TROPOMI pixel size is located at the edge of the swath and has grown to 20-25 km in the across-track direction, are not a good representation of the ground measurements.

Limiting the analysis to nadir-views and usage of the mRPV model (i.e. the most favourable conditions) could provide stronger constraints. Five orbits have overpasses with viewing zenith angles less than 10 degrees, although none share a common orientation. Two of the comparisons (for orbit 15272 and 17996) show ratios with values above 1.2. The other three have ratios close to unity. Interestingly, the two large deviations are measurements taken in around the winter of 2020/2021 and show almost perfect agreement between MODIS, VIIRS and the mRPV model, a property not shared between many of the other overpasses. As such, data on nadir-views remains inconclusive.

For TROPOMI-SWIR, ground measurements with the data quality as obtained for RRV but using a significantly larger homogeneous site (e.g. the Saharan desert sites as used in Bacour et al. (2019) and van Kempen et al. (2021)) would likely provide an improvement to the results. However, due to its capability to produce daily global coverage and thus its varying pixel size and orientation over reference sites, TROPOMI-SWIR will remain facing unique challenges that are not faced with





equivalent sensors such as GOSAT, GOSAT-2, OCO-2 and OCO-3. In addition, large viewing angles are poorly represented by MODIS and VIIRS BRDF products. Although the mRPV model appears to provide an improvement, this method still has limitations. Despite the limitations found in our analysis, we realize that the RRV remains one of the easiest accessible and most well-maintained sites on the planet to provide ground-based measurements from which validation of radiance can be done. Larger invariant sites face numerous geo-political and natural challenges.

### 6.4 Suggestions for other sounders

Conclusions from the discussion above contain important recommendations for future missions on vicarious calibration. The most important is the size of a pixel, its orientation and the zenith angle at which a reference site such as RRV is viewed. Multiple consistent measurements using a small (i.e. $1\times1$ km$^2$) field of view at a small ($<10\deg$) zenith angle with identical orientations are expected to severely limit the uncertainties and/or variations seen in the analysis of this work. The assumption of homogeneity would be much more applicable then TROPOMI-SWIR. Such measurements would also have a small (i.e. $<3\%$) BRDF correction factor. More importantly, this factor will vary less from measurement to measurement when using the correction derived using the mRPV model. Although some variations are expected, as seen from the MISR data, small variations will make stronger constraints possible. Whether or not such constraints also arise from the usage of the VIIRS BRDF cannot be derived from our data, but the expectation is that such data products are more representative of the physical location than of TROPOMI-SWIR.

### 7 Conclusions

The conclusions of this work can be summarized as follows.

– The TROPOMI-SWIR module is still very stable after five years in orbit, as evidenced by the trend monitoring of the solar irradiance and white light source. This corroborates and extends the conclusions of van Kempen et al. (2019), Ludewig et al. (2020) and van Kempen et al. (2021). Currently, the degradation is quantified to a 0.15% loss in transmission and a 0.3% loss in the number of usable detector pixels.

– The vicarious calibration methodology validates the results from the internal calibration unit. However, the calibration unit provides far superior results. Vicarious calibration limits are an order of magnitude larger (4-10%).

– As such, the decision to use a calibrator unit for TROPOMI-SWIR module monitoring has proven to be very prudent.

– The comparison between the RADCALNET data provides an upper limit of $\sim8\%$ to the amount of instrument degradation. The method does show that non-Lambertian corrections are necessary, as evidenced by the systematics in the residuals of the two models.

– The RRV ground calibration campaigns provide an independent method to verify the stability of the TROPOMI-SWIR module, although the accuracy is relatively poor ($\sim6$ to 10%), compared to the result from internal calibration sources.



We explored three correction methods using (i) an mRPV model, (ii) a BRDF model using MODIS data, and (iii) a BRDF model using VIIRS data. The mRPV method produces superior results. In particular measurements at large ($>50\deg$) viewing angles are poorly represented by VIIRS and MODIS data.

- Vicarious calibration of TROPOMI is severely limited by the relatively large pixel size and varying pixel orientation concerning the RRV location. However, the method is one of the few that can truly independently verify instrument stability at radiance level. Similarly, RRV provides one of the few sites on the planet with sufficient ground measurements to perform vicarious calibration.

- Many of the limitations found in this work for TROPOMI-SWIR are expected to be strongly reduced for instruments with smaller pixel sizes using uniform orientations with soundings close to the nadir.

*Data availability.* The radiance of Sentinel-5p TROPOMI-SWIR is publicly available at the pre-operations science Hub. RADCALNET is publicly available, on request, at www.radcalnet.org.

*Competing interests.* There are no known competing interests

*Author contributions.* TvK and TR carried out the analysis, and TvK wrote the manuscript. RvH is responsible for the monitoring system used for the TROPOMI monitoring. CB, RR, TP, and DF supplied the data taken at RRV in the dedicated campaigns. RH and IA proofread the final document.

*Acknowledgements.* This paper contains Copernicus Sentinel data. The research of TvK is funded by the TROPOMI national program from the Netherlands Space Office (NSO). We would like to thank the hard work of people who contributed to the data acquisition at RailRoad Valley, particularly Mark Helmlinger, Linley Kroll and Fumie Kataoka.



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
