# Peer review of "Vicarious Calibration of the TROPOMI-SWIR module over the Railroad Valley playa"

_EGUsphere, 2023_

## Author Response (AR1)

REPLY TO REFEREE 1

>>
Dear referee,

Thank you for the kind words. Please find below responses that complement the changes that were put in the new version of the paper. Indeed, the usage of onboard calibration should be highlighted even more.
I also want to apologise for the very long delay in this reaction, and subsequently thank you for your patience. Due to the absence of a second referee, this response has been sitting on my harddrive a long time.

with kind regards,

Tim

=====================
Abstract:    - You speak about vicarious calibration, but is it actually calibration, i.e. are correction factors derived and applied, or is it only verification?

> Indeed, as applied to TROPOMI-SWIR, is it only a verification. Although correction factors could have been derived (with large uncertainties), these are never applied to TROPOMI-SWIR ready. However, to introduce confusing differences with existing literature in this field, we elected to keep the term 'Vicarious Calibration'

  - The last sentence about added value in the vicarious calibration w.r.t. the onboard calibration is very vague. Probably because your end conclusion is actually rather the opposite: that there is hardly any added value (if there are on-board calibration means)?

> We added an additional sentence. In our opinion there is added value. Despite the very large uncertainties, vicarious calibration remains an independent verification. In addition, it also reveals how design choices of new instruments must be made to make more effective use of vicarious calibration.

Introduction:    - line 11: perhaps specify that this is about the SWIR detector as the numbers (in particular the resolution on the ground) are different for the UVN module.
> corrected
  - some unexplained acronyms (RADCALNET, JPL LED), also happens in other locations in the paper.

> corrected

Section 2.1:    - line 1: from on-ground measurements -> from pre-flight on-ground measurements

> corrected
  - Transmission: this is meant as both transmission along the optical path and detector sensitivity?

> This was not our intent, but effectively, you are correct. This is due to the ordering of sections 2.1, 2.2 and 2.3. By changing the order (2.1 is now the last), this ambiguity has been removed. The transmission should only aim towards the optical path. The detector sensitivity is in part covered by the dark flux and degradation.

Section 2.3:    - line 1 on p8: pixels -> pixel's?

> corrected

Section 3.1:    - is this poorer spectral resolution of RADCALNET (10nm) still fine enough to really probe only the continuum you're aiming for?

> No it is not. This is discussed in 4.4.1. Apart from the broad response of 10 nm, the shape of the response (triangular) also poses issues.

Section 3.2:    - You write: "This variation is dominated by the varying viewing angles, represented by the Bidirectional Reflection Distribution Function (BRDF) of the    non-Lambertian desert surface." How do you know? Is there a strong correlation with viewing angles but not solar angles or e.g., aerosol load?

> First, it should be that it is dominated by varying viewing AND solar angles. Thank you for catching this error.
This is an effective conclusion from the pair of papers from Bruegge et al., 2019a and 2019b. Although aerosol load could play a major role, the effective Aerosol optical depth (AOD) at 2.3 micron above RRV is very low. Even with large AOD factors in the visible, the term remains relatively neglible at 2.3 micron.

Section 4.1.2:    - Aerosols are not considered in RemoteC?

> yes it is. Changed the text to reflect it.

Section 4.1.3:    - Please justify the use of a linear extrapolation (I guess this is what you expect for Mie scattering at these wavelengths)?

> changed

Section 4.2.1:    - I'm not sure the full mathematical description is of the model is needed here, but if you provide it, it would be nice to have some description of what the different parameters physically mean (e.g. r_0,k,b), if they can be given an intuitive meaning.

> this is explained in Bruegge et al., 2019a, but more thoroughly in the Rahman papers from 1993. Conceptually it is not easy to relate the parameters to physical phenomena.

Section 4.3.2:    - What causes the remaining variability, besides the SZA evolution?

> unknown. we hypothesized even small variations in elevations (or even surface 'roughness') causes small-scale shadowing. But there is no evidence to support this hypothesis, so it was decided to leave this out of the paper. Similarly, we discussed whether or not a very small component of the total emission is not due to the reflection, but due to thermal emission of a hot surface. at 300-320 K, blackbody radiation at 2.3 micron is nonzero. But we have no surface temperature measurements to confirm/disprove this.

Section 5.1:    - Fig. 9: I don't think I understand the units of the y-axis, the range seems huge for a multiplicative correction factor. How is this correction factor applied?

> This is indeed a multiplicative factor that is applied on the raw spectrum. This is never shown as the resulting spectrum are the blue spectra in Fig. 11. Some extreme values were not applied, although some extreme values indeed show a difference of almost an order of magnitude in the resulting comparison similar to Fig. 11 (i.e. a simulated spectrum would be 7 times brighter than the observed).
    - General comment on figures throughout the paper: many labels are rather small. E.g., in Figure 12.

> We tried changing the fonts and font sizes, but were relatively unhappy with the results.

Section 6.2.1:    - Maybe I missed it, but would it not be an interesting exercise to apply this (probably too strict) TROPOMI CH4 cloud filter? You'll lose a lot of data, but at least cloud contamination should be minimal.

> Yes, this was done, with relatively poor results. Due to the high albedo in general, the fits of CH4 above whiter deserts such as RRV (and similarly salt flats) do not always converge, even in known cloudless conditions. You indeed get very very few reliable points (10 per year)

Section 6.3:    - The first sentence sounds a bit overconfident. I guess you mean "For TROPOMI-SWIR, vicarious calibration cannot be improved upon from the RRV analysis presented in this work."

> changed

  - 1st paragraph: so a more detailed mapping of the BRDF over the entire RRV (allowing you to drop the homogeneity assumption) would be a great step forwards? Then again, looking at Figure 2, it seems many pixels even reach outside the valley itself.
> Indeed, after submission we looked at this using VIIRS BRDF data interpolated/modelled, but did not get an improvement.

  - At some point, true horizontal sensitivity over the pixel will probably also be an issue (i.e., it not being top-hat like but rather a super-Gaussian of some sort?).

> very likely the spatial response function (i.e. the horizontal sensitivity) already plays a role. But we are unable to assess how big. We theorized that the systematic increase seen in the median of the mRPV model comparisons of 2-3 % (Table 4) might be due to this effect. However, with the larger uncertainties of all the other effects, we cannot, and likely will never be able to, discern between statistical effects and the spatial response of a pixel.

REPLY TO REFEREE 2

>> Dear Referee,

Thank you for the kind words and the suggestions. We have adopted them in the new version of the paper that has been submitted. Specific comments to specific suggestions are given below. If no comment is given, we have adopted your comments in full.

with kind regards,

Tim van Kempen
* * *
Abstract

Page 1, Line 7: I would suggest to give a short description of the location Railroad Valley Playa in the abstract. Currently, there is only reference to the region but why this is special and appropriate region for vicarious calibration of TROPOMI-SWIR module is missing. I think this is something that the reader wants to know already when reading the abstract as it is mentioned in the title of this manuscript.

> included in the abstract.

Page 1, Line 10: I would suggest to provide references for the vicarious calibration of OCO-2 and GOSAT missions in the abstract.

> The guidelines of the AMT journal dictates that references should not appear in the abstract. They are given in the introduction (Kuze et al., 2014, Bruegge et al., 2019a, 2019b and 2021)

Section 1 (Introduction)

Page 2, Line 12: What happened with the 25 columns and 39 rows? What do the authors mean with the statement "about 975 columns and 217 rows are illuminated"?

> The columns are not all illuminated due to a cover that is on the detector. The rows are not all illuminated due to the exit angle of the dispersing element.

Section 2 (TROPOMI-SWIR Performance)

Page 4, Line 5: The change in the integration time should have been effective since August 2019 and not 2020. Please cross-check this important date. Moreover, I would propose to rephrase the sentence as "effectively changing the spatial resolution in the along-track direction from …".

> thank you, this was indeed a typo.

Page 4, Line 13: The authors refer to the instrument zenith angle (i.e., distance from the nadir pixel) and just one line above they refer to the viewing zenith angles. What is the difference that the authors imply between the two zenith angles? I would recommend them to give the definitions.

> these are the same, clarified.

Section 2.1 (Transmission stability), Section 2.2. (Dark Flux)

> We would like to inform the referee that the order of sections 2.1, 2.2 and 2.3 has changed for clarity.

Section 3.2 (RRV campaigns)

Page 9, Line 28: I would change to "an area of 500 x 500 m2" instead of "500 meters by 500 meters". In general, the sentence "The field measurements of .... TROPOMI at nadir" is a bit too long and complicated for me to follow it well.

> rephrased

Page 10, Line 1: JPL is an acronym which is not introduced in the text.

> in the current version it is used in the introduction.

Section 4.2.1 (mRPV)

Page 14, Figure 6: I cannot really see any pattern or a seasonal variation for the free parameters n the mRPV model. This figure is not so informative from my point-of-view.

> the main pattern is the correlation between the three parameters and the visualisation that the r_0, k and b parameters are not static for the desert surface.

Page 14, Lines 4-14: I would probably move all the mathematical formulas in an appendix. As far as I understand, the formulas are not originally derived for this study. Therefore, the reference to the source and the description in an appendix would be more optimal for me. Moreover, there are no explanations about the free parameters r0, k and b. What do they represent? What does "hot spot" actually mean?

> This was discussed internally before submission, but decided against. The main motivation is that the comparison between the mRPV model and VIIRS/MODIS, including the different corrections, are a central point to the paper. It is our opinion that the lessons learned of the methodology as applied to the data is a more important conclusion that the final percentages derived from it. As such putting the mathematical formulae within the paper seemed prudent. During writing we kept going back to these often to understand how these relate to the corrections.
The relation of the free parameters themselves, as well as the hot spot, as applied to RRV are extensively discussed in Bruegge et al., 2019a, 2019b, in addition to the original papers from Rahman et al., 1993a, 1993b. These papers are also referenced. As opposed to the formulae, we feel that the references should be sufficient.

Section 4.3.1 (Spatial Averaging)

Page 16, Line 7: By how much is the error? An estimate should be given.

> we do not know, but discuss this effect in the Section 6. In fact most of the uncertainty in the final result of this paper likely is related to the large pixel size and the spatial averaging

steps. We rephrased the sentence to remove the ambiguity and refer to the uncertainties seen in earlier studies.

Section 4.3.2 (Time differences and Solar angle)

Page 16, Line 9: "Variations also exist as a function of the time of day." What do the authors mean with this statement? It is not clear to me.

> sentence remove the ambiguity.

Section 4.4.1 (RADCALNET)

Page 17, Line 19: I didn't understand how this multiplication factor of 1.247 was derived.

> It is the average difference between all dates which have a cloudless ToA RADCALNET radiance at 2310 nm and a the 2313 continuum radiance seen by TROPOMI where the viewing angle is less than 3 degrees (i.e. nearly nadir-viewing). For such very small degrees, there is assumed to be no correction nBRDF needed. The difference is solely due to the absorption of methane and water in the RADCALNET bandwidth (~10 nm width triangular)

Section 4.4.2 (Ground campaigns)

This section seems incomplete. Could the authors elaborate more on how the radiances are derived from the reflectances?

> This is described in 4.1.2. RemoteC is used to calculate the radiative transfer of irradiance values through reflection on the RRV surface back to the TROPOMI entry.

Page 19, Lines 5-8: I would like to ask the authors to comment further on the uncertainties which are introduced due to the assumption of spectral dependence absence.

> added. This uncertainty is assumed to be very minor.

Section 5 (Results)

Page 20, Line 13: Elaborate more on the statement "the accuracy of the mRPV model". How should the reader interpret this? mRPV model is not accurate enough at larger VZA?

> Yes. The model was build upon many measurements of the Earth in the 90s, none of which had angles larger than 40-45 degrees at the time. We do note that the model significantly outperforms both the MODIS and VIIRS data products at these larger angles.

Section 5.2 (TROPOMI-SWIR vs RadCalnet)

Page 22, Line 2: "Structural trends in the residuals are seen for the Lambertian model, but not the mRPV model." I cannot see these trends. I would ask the authors to elaborate more on it.

> these trends are seen in the residuals of Fig. 10. Here the Lambertian model (top) shows slopes in time (e.g. second half of 2020 and first half of 2021). The mRPV residuals are random (i.e. a fitted slope would be 0)

Section 5.3 (TROPOMI-SWIR vs dedicated campaigns)

Page 23, Line 5: Where is MIPREP acronym defined?

> rephrashed/removed the MIPREP reference.

Page 24, Figure 12: There is a small mistake here. Please replace "VIIRS (left)" with "VIIRS (right)" in the caption. I would also like to see a statement related to the second sentence in the caption. It's OK to omit ratios above 1.5 but it would be informative to specify how often those ratios were found. I would recommend to give a percentage for MODIS and VIIRS separately.

> Corrected. We do not give a percentage, but an absolute number, mirroring the text.

Section 6.3 (Implications for TROPOMI)

Page 27, Line 29: "As such, data on nadir-views remains inconclusive". Probably more orbits with nadir-views should be investigated. A set of 5 orbits is a very limited sampling to draw any conclusions. Moreover, the word "data" is plural and there is a grammar issue; please correct "remains" to "remain".

> I fully agree. The mostly manual data acquisition is optimized for OCO and GOSAT which have a much poorer coverage.

Section 7 (Conclusions)

Page 28, Line 24: "Vicarious calibration limits are an order of magnitude larger (4-10%)." It is not clear to me what do the authors mean with this statement and to which findings/comparisons they refer to?

> it is to the on-board results in the previous bullet. Rephrased for clarity.